# Enhanced Stability and Driving Performance of GO–Ag-NW-based Ionic Electroactive Polymer Actuators with Triton X-100-PEDOT:PSS Nanofibrils

**DOI:** 10.3390/polym11050906

**Published:** 2019-05-19

**Authors:** Minjeong Park, Seokju Yoo, Yunkyeong Bae, Seonpil Kim, Minhyon Jeon

**Affiliations:** 1Department of Nanoscience and Engineering, Center for Nano Manufacturing, Inje University, Gimhae 50834, Korea; mjpark9121@gmail.com (M.P.); yuinjae7@gmail.com (S.Y.); yunkyeong.b@gmail.com (Y.B.); 2Department of Military Information Science, Gyeongju university, Gyeongju 38065, Korea; seonpil@gu.ac.kr

**Keywords:** graphene oxide, silver nanowires, ionic electroactive polymer, poly(3,4-ethylenedioxythiophene)–poly(styrenesulfonate) (PEDOT:PSS), 4-(1,1,3,3-Tetramethylbutyl)phenyl-polyethylene glycol

## Abstract

Ionic electroactive polymers (IEAPs) have received considerable attention for their flexibility, lightweight composition, large displacement, and low-voltage activation. Recently, many metal–nonmetal composite electrodes have been actively studied. Specifically, graphene oxide–silver nanowire (GO–Ag NW) composite electrodes offer advantages among IEAPs with metal–nonmetal composite electrodes. However, GO–Ag NW composite electrodes still show a decrease in displacement owing to low stability and durability during driving. Therefore, the durability and stability of the IEAPs with metal–nonmetal composite electrodes must be improved. One way to improve the device durability is coating the electrode surface with a protective layer. This layer must have enough flexibility and suitable electrical properties such that it does not hinder the IEAPs’ driving. Herein, a poly(3,4-ethylenedioxythiophene)–poly(styrenesulfonate) (PEDOT:PSS) protective layer and 4-(1,1,3,3-tetramethylbutyl)phenyl-polyethylene glycol (Triton X-100) are applied to improve driving performance. Triton X-100 is a nonionic surfactant that transforms the PEDOT:PSS capsule into a nanofibril structure. In this study, a mixed Triton X-100/PEDOT:PSS protective layer at an optimum weight ratio was coated onto the GO–Ag NW composite-electrode-based IEAPs under various conditions. The IEAP actuators based on GO–Ag NW composite electrodes with a protective layer of PEDOT:PSS treated with Triton X-100 showed the best stability and durability.

## 1. Introduction

Ionic electroactive polymers (IEAPs) are among the most functional materials-based actuators. IEAPs have useful properties, such as a lightweight composition, a large working displacement under a low driving voltage, and a high energy density [1]. Electrodes are an important component of IEAPs. When sufficient, they provide high electrical conductivity, mechanical durability, and a smooth surface morphology [2]. The electrodes of IEAPs are categorized as metallic or nonmetallic. Noble metals (e.g., platinum, gold) with high electrical conductivity and electrochemical stability are often used as the metallic electrode of IEAPs. However, most metallic electrodes exhibit microcracks on the electrode surface, thus diminishing their surface electrical conductivity during long-term actuation [3,4]. Meanwhile, most nonmetallic electrodes are fabricated using transition metal oxides or carbon materials [1,5,6,7,8,9]. These materials can be assembled into electrodes via a physical hot-pressing method, which is simpler and faster than other fabrication methods [10,11]. However, nonmetallic electrodes are less conductive than metallic electrodes. To resolve such challenges, recent studies have increasingly investigated metallic–nonmetallic composite electrodes. Composite electrodes can solve both the occurrence of microcracks on the metallic electrode surface during driving and the low electrical conductivity of nonmetallic electrodes [12]. However, composite electrodes have a high contact resistance on their surfaces, which can instigate electrode burn out and decrease electrode durability during actuation.

Poly(3,4-ethylenedioxythiophene)–poly(styrenesulfonate) (PEDOT:PSS), a conductive polymer, is applied in various fields for its advantages of electrical conductivity and transparency. Particularly, PEDOT:PSS has been actively studied as a protective layer coated onto electrode materials such as carbon nanotubes (CNTs), graphene, and metal nanowires (NWs) with high contact resistance [13,14,15,16]. PEDOT:PSS is composed of a PEDOT phase and a PSS phase. The PEDOT phase is electrically conductive, but it is not suitable for solution processing because of its low solubility. On the other hand, the PSS phase is electrically insulating and water soluble. Because of these combined properties, the PEDOT:PSS is both electrically conductive and water soluble. However, the PEDOT phase’s tendency to aggregate can make electrodes brittle.

Triton X-100 (4-(1,1,3,3-tetramethylbutyl)phenyl-polyethylene glycol) is composed of hydrophilic polyethylene oxide and hydrophobic 4-(1,1,3,3-tetramethylbutyl)-phenyl. This composition imparts it with amphiphilic properties owing to its hydrophilic “head” and hydrophobic “tail” and is thus a nonionic surfactant. It is widely used for mixing polar and nonpolar materials or lysing cells to extract proteins or organelles in biological fields through its surfactant property [17]. Moreover, Triton X-100 can lead to the formation of PEDOT nanofibrils in a viscoelastic medium due to its amphiphilic molecular structure of Triton X-100. Additionally, it can solve the PEDOT phase aggregation problem. Through this mechanism, the structure of PEDOT:PSS can be modified to have a flexible morphology. In addition, it enables improving the electrical conductivity by removing the PSS phase, which acts an insulator, through post-treatment [18,19,20].

In this study, Triton X-100-PEDOT:PSS-coated graphene oxide and silver NWs (TP/GO–Ag NWs) were fabricated as composite-electrode-based IEAPs. Herein, it is demonstrated that Triton X-100 can transform the aggregated structure of PEDOT:PSS into a nanofibril structure and thus improve electrical properties. Furthermore, the Triton X-100/PEDOT (TP) mixture coated on GO–Ag NW composite electrodes is used as a protective layer for the GO–Ag NW electrode. The improved characteristics of TP/GO–Ag-NW-based IEAPs were observed with a focus on electrical conductivity and driving properties.

## 2. Materials and Methods

### 2.1. Materials

PEDOT:PSS, Triton X-100, GO, and Ag NWs were used to fabricate the composite electrodes of the IEAPs. PEDOT:PSS (1.3 wt %) and Triton X-100 were purchased from Sigma-Aldrich, St. Louis, MO, US. GO was synthesized using Hummer’s method [21]. The Ag NW solution was purchased from Duksan Hi-Metal, Ulsan, Korea. Nafion 117 (N117) and 20 wt % Nafion resin (Nafion solution) were purchased from the DuPont Company, Midland, MI, US. 1-Ethyl-3-methylimidazolium trifluoromethylsulfonate (EMIM-Otf), an ionic liquid (IL), was purchased from Merck KGaA, Darmstadt, Germany. A polyvinylidene difluoride (PVDF) membrane filter with a pore size of 0.20 μm and a diameter ∅ 47 mm was purchased from Hyundai Micro., Ltd, Seongnam, Korea.

### 2.2. Fabrication of IEAP Actuators Based on TP/GO–Ag NW Electrode

GO powder (330 mg) and deionized water (5 mL) were stirred at 250 rpm for 5 min in a beaker to prepare the GO solution. The GO and Ag NW solutions were mixed in a 1:2.5 volume ratio. This GO–Ag NW mixture was used to fabricate a composite paper electrode using a vacuum filtration system. This paper electrode was dried at 100 ℃ for 5 min in a vacuum oven. We performed ion exchange on the purchased Nafion following a previously-reported method [19]. The Nafion resin was painted directly onto both surfaces of the N117 membrane as an additive solution to paste the GO–Ag NW composite electrode to the Nafion membrane, forming an electrode/membrane/electrode structure. This structure was then hot-pressed at 0.1 MPa and 100 °C for 2 min. Thus, we obtained IEAPs based on GO–Ag NW electrodes. In addition, Triton X-100 was mixed with PEDOT:PSS at various volume ratios and spin-coated onto both surfaces of the IEAP actuator based on the GO–Ag NW composite electrodes. Two coating conditions were considered: duration and speed. The PSS phases were then removed with methanol. Finally, IEAPs based on TP/GO–Ag NW composite electrodes were fabricated (width × length (Free length) × thickness = 5 mm × 35 mm (30 mm) × 200 μm).

### 2.3. Characterization

The transmittances of samples with different Triton X-100 weight ratios were investigated using an ultraviolet visible spectrophotometer (LAMBDA 465, PerkinElmer, Seoul, Korea). The morphologies and thickness of the electrodes were investigated using atomic force microscope (AFM; XE-100, Park Systems, Suwon, Korea) and field-emission scanning electron microscopy (FE-SEM; S-4300, Hitachi, Tokyo, Japan). The sheet resistances of electrodes were measured using a four-point probe (FPP-HS 8, DASOL ENG, Cheongju, Korea). The driving characteristics of the actuators were measured using a laser displacement sensor (ZS-LD80, OMRON Korea, Seoul, Korea) in an actuation performance analyzer that we constructed.

## 3. Results

### 3.1. Characterization of the TP/GO–Ag NW Electrode

Triton X-100 and PEDOT:PSS were mixed at six different weight ratios (0.0, 1.0, 2.5, 5.0, 7.5, and 10.0 wt % Triton X-100). In addition, the PSS phases of all Triton X-100 and PEDOT:PSS mixtures were removed with methanol. In order to optimize the Triton X-100 concentration in the TP mixtures, protective layers were separately spin-coated on glass. Figure 1a shows the TP mixtures with different Triton X-100 contents coated on glass. Pure PEDOT:PSS did not uniformly coat the glass because of the high surface tension of PEDOT:PSS and its aggregated structure. However, the added Triton X-100 reduced the surface tension of the various TP mixtures. Thus, the TP mixtures uniformly coated the glass, in contrast with the pure PEDOT:PSS solution. 

Figure 1b shows the optical transmittance at 550 nm and the thickness of the pure PEDOT:PSS and TP mixtures coated on the glass. The transmittance and thickness of all samples are inversely proportional. Pure PEDOT:PSS had lower transmittance (84.29%) than the TP mixtures. Meanwhile, the TP mixture with 7.5 wt % Triton X-100 had the highest transmittance (88.81%) and the thickest coating (164.29 nm) among the TP mixtures. These results suggests that the aggregated structure of PEDOT:PSS transformed into a nanofibril structure through the addition of Triton X-100.

Figure 2a–f shows AFM images of the 0.0, 1.0, 2.5, 5.0, 7.5, and 10.0 wt % TP layers, respectively, which reveal the surface roughness and structure of the TP layer. For accurate comparison, we scanned and analyzed an area of 5 × 5 μm^2^. These results reveal that the Triton X-100 treatment transformed the PEDOT:PSS aggregated structure to a nanofibril structure. Notably, the TP mixture with 7.5 wt % Triton X-100 had a nanofibril structure and lower surface roughness than the other TP mixtures, thus corroborating the transmittance and thickness results. At 10.0 wt %, the of the TP mixture exhibited aggregation, meaning that the amount of Triton X100 exceeded the critical micelle concentration (CMC), thereby forming micelles. The CMC, the surfactant concentration above which micelles form, is an important characteristic of a surfactant. Specifically, after reaching the CMC, any additional surfactants added to the system form micelles, which are large molecules formed by clusters of surfactant particles such as Triton X100. The size of the particles increases with increasing molecular aggregation. Figure 2g schematically illustrates the mechanism underlying the PEDOT:PSS transformation with the Triton X100 treatment.

Figure 3 graphically represents the sheet resistance of the GO–Ag NW composite electrodes coated with different TP mixtures, which was measured using a four-point probe. The TP-mixture-coated GO–Ag NW electrodes had a lower sheet resistance than the GO–Ag NW electrode without Triton X-100. In addition, washing with methanol decreased the sheet resistances of all the TP-mixture-coated GO–Ag NW electrodes. Particularly, the 7.5 wt % Triton X-100 TP mixture had the lowest sheet resistance. Table 1 lists the sheet resistance of the samples in detail. Accordingly, the 7.5 wt % Triton X-100 TP mixture is the most suitable choice as the protective layer for IEAP electrodes.

Next, to optimize the spin coating conditions of the TP mixture with 7.5 wt % for the GO–Ag NW electrode, the TP/GO–Ag NW electrodes were fabricated with different coating times (0, 15, 30, and 45 s) and coating speeds (300, 500, 700, 1000, and 2000 rpm). Figure 4a shows the sheet resistance of the TP/GO–Ag NW electrodes with different coating times and a fixed coating speed of 1000 rpm. In this figure, the results for the TP-mixture-coated electrodes are compared with those of the GO–Ag NW electrode and the GO–Ag NW electrode coated with the pure PEDOT:PSS (P/GO–Ag NW electrode). The TP mixture with 7.5 wt % Triton X-100 coated for 30 s had a lower sheet resistance than the other samples. Figure 4b shows the sheet resistance of the TP/GO–Ag NW electrode at different coating speeds (coating time: 30 s). The 7.5 wt % Triton X-100 TP mixture coated at 1000 rpm for 30 s provided the lowest sheet resistance of about 161 mΩ/sq. Thus, the optimal spin coating conditions for coating the TP mixture on the GO–Ag NW electrode were 7.5 wt % Triton X100, a coating time of 30 s, and a coating speed of 1000 rpm. The sheet resistance of this electrode was 49.73% and 82.10% lower than those of the GO–Ag NW electrode and the P/GO–Ag NW electrode, respectively. These optimized conditions were thus used for subsequent experiments.

### 3.2. Actuation Performance of IEAP Actuators Based on TP/GO–Ag NW Electrode

The driving performance of IEAP actuators with different electrodes was measured and observed, and the results are presented in Figure 5. The PEDOT:PSS-based IEAPs were measured to confirm the effect of 7.5 wt % Triton X-100. In order to investigate the effects of the electrode type on each actuator, the actuation performances of the IEAPs with three types of electrodes (GO–Ag NWs, P/GO–Ag NWs, and TP/GO–Ag NWs) were measured under ± 2.5 V_AC_ and 0.2 Hz, as shown in Figure 5a,b. Figure 5a shows the harmonic responses of the three types of IEAPs. The TP/GO–Ag-NW-based IEAPs had larger tip displacements than the other IEAPs. 

As shown in Figure 5b, the peak-to-peak performance of the TP/GO–Ag-NW-based IEAPs showed a lower slope than that of the other IEAPs, meaning that the TP/GO–Ag-NW-based IEAPs are more durable than the other IEAPs. Figure 5c shows the actuation performance, from 0 s to 1.3 s, of a segment of the harmonic response from Figure 5a. The response rate of the TP/GO–Ag-NW-based IEAPs was 34.83% and 23.87% faster than those with the P/GO–Ag-NW-based IEAPs and GO–Ag-NW-based IEAPs, respectively. Figure 5d shows the curvatures of the three types IEAPs. The maximum curvature of TP/GO–Ag-NW-based IEAPs was approximately 1.054 m^−1^, which is higher than that of the GO–Ag-NW-based IEAPs (0.858 m^−1^) and P/GO–Ag-NW-based IEAPs (0.53 m^−1^).

After measuring the actuation performance, SEM was used to observe the change in the electrode surfaces of the three types IEAPs. The decreased actuation performance of the GO–Ag-NW- and P/GO–Ag-NW-based IEAPs in Figure 5a,b can be explained by Figure 6, which shows SEM images of the electrode surface of three types of IEAPs before (Figure 6a–c) and after (Figure 6d–f) the driving test. Because of the high contact resistance of the surface, heat is generated, which weakens metal NWs. This is a critical drawback of metal NWs networks, which may disconnect when voltage is administered to IEAPs owing to the resulting heat. Accordingly, serious transformation and network disconnection was observed for the Ag NWs in the IEAPs based on GO–Ag NWs and P/GO–Ag NWs. In contrast, the shape and network connection of the Ag NWs in the TP/GO–Ag-NW-based IEAPs were well maintained. Ultimately, this TP layer, which shows enhanced stability and durability during driving, can be used as a protective layer to decrease the high contact resistance of the GO–Ag NW electrode.

## 4. Discussion and Conclusions

Triton X-100, a nonionic surfactant, was used to functionally enhance a PEDOT:PSS protective layer on a GO–Ag NW electrode. Triton X-100 induced the shape deformation of PEDOT:PSS, which reduced both sheet resistance and surface tension. When applied to the GO–Ag NW electrode, the PEDOT:PSS mixed with 7.5 wt % Triton X-100 provided the lowest sheet resistance. The optimal coating conditions for PEDOT:PSS mixed with 7.5 wt % Triton X-100 were 30 s of coating at 1000 rpm. The sheet resistance of the TP/GO–Ag NW electrode coated under these optimal conditions was 160 mΩ/sq., which was 49.73% and 82.10% lower than those of the GO–Ag NW and pure PEDOT:PSS coated GO–Ag NWs (P/GO–Ag NWs) electrodes, respectively. The driving performance of TP/GO–Ag-NW-based IEAPs was significantly better than that of the IEAPs based on GO–Ag NWs and P/GO–Ag NWs. Furthermore, the shape and network connection of the Ag NWs in the TP/GO–Ag-NW-based IEAPs was well maintained, as revealed by SEM images. Therefore, both the stability and durability of TP/GO–Ag-NW-based IEAPs were confirmed to improve. These results demonstrate the possibility of improving electrodes with high contact resistance in terms of durability and stability.

## Figures and Tables

**Figure 1 polymers-11-00906-f001:**
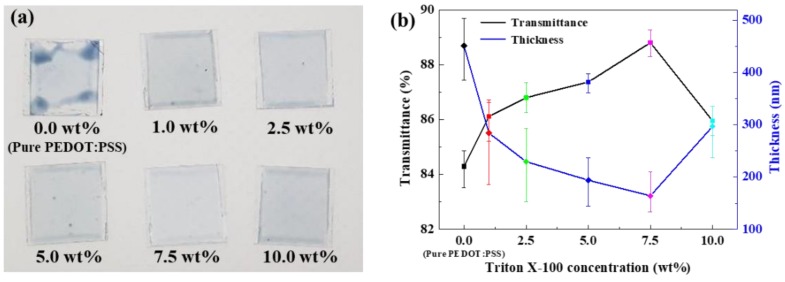
(**a**) The optical images of Triton X-100/poly(3,4-ethylenedioxythiophene) (TP) mixture coated on glass and (**b**) transmittance (black line) and thickness (blue line) of films with different Triton X-100 weight ratios.

**Figure 2 polymers-11-00906-f002:**
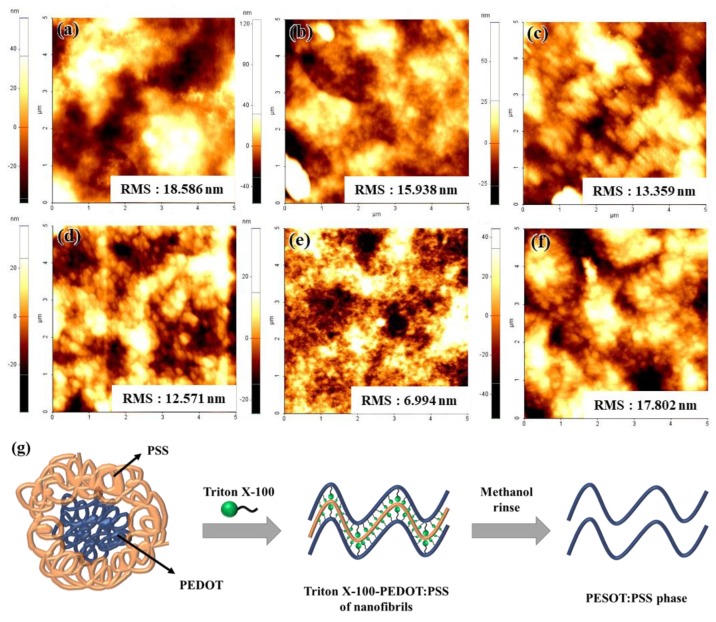
Atomic force microscope (AFM) images with a scanning area 5 μm × 5 μm and root-mean-square (RMS) surface roughness values. (**a**) 0.0 wt %, (**b**) 1.0 wt %, (**c**) 2.5 wt %, (**d**) 5.0 wt %, (**e**) 7.5 wt %, (**f**) 10.0 wt % Triton X100. (g) Schematic diagram illustrating the phase transition of poly(3,4-ethylenedioxythiophene)–poly(styrenesulfonate) (PEDOT:PSS) in the presence of Triton X-100.

**Figure 3 polymers-11-00906-f003:**
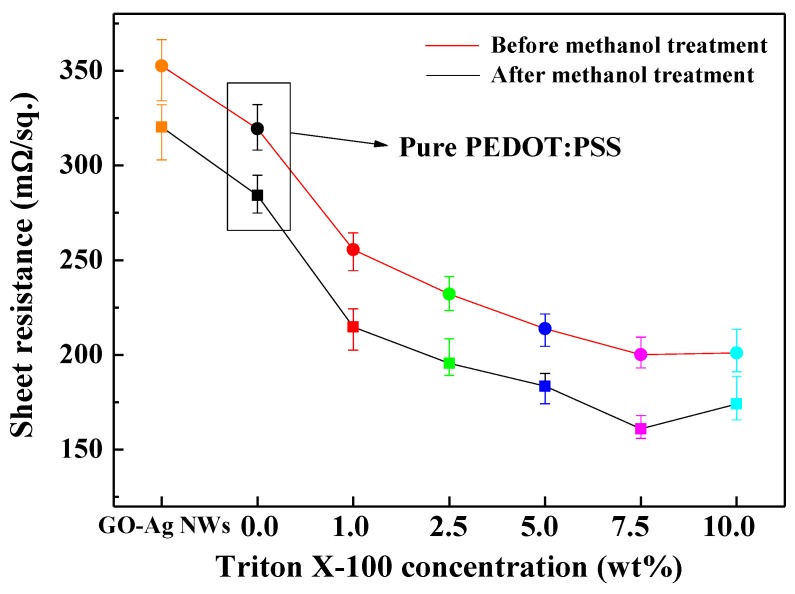
Sheet resistances before (red line) and after (black line) methanol treatment with various Triton X-100 weight ratios.

**Figure 4 polymers-11-00906-f004:**
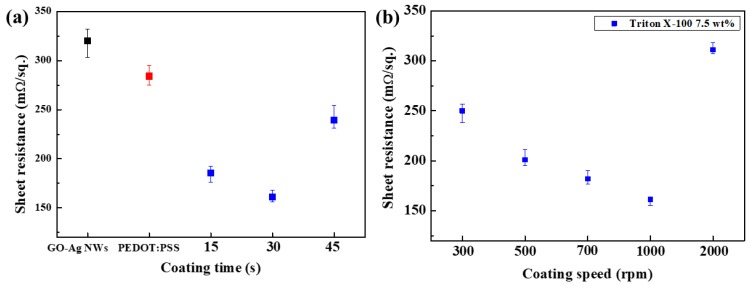
Sheet resistances of the (**a**) uncoated electrode (black point), P/GO–Ag NW electrode, and 7.5 wt % Triton X100 TP/GO–Ag NW electrodes coated for different coating times at 1000 rpm. (**b**) Sheet resistances of the 7.5 wt % Triton X100 TP/GO–Ag NW electrodes coated at different coating speeds.

**Figure 5 polymers-11-00906-f005:**
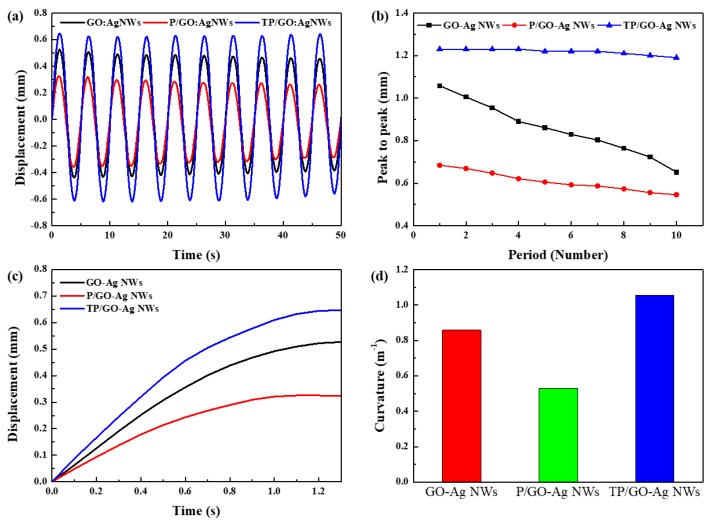
(**a**) Displacement versus time (± 2.5 V_AC_, 0.2 Hz), (**b**) peak-to-peak performance, (**c**) response rate, and (**d**) bending curvature of three different ionic electroactive polymers (IEAPs) (based on GO–Ag NWs, P/GO–Ag NWs, and TP/GO–Ag NWs).

**Figure 6 polymers-11-00906-f006:**
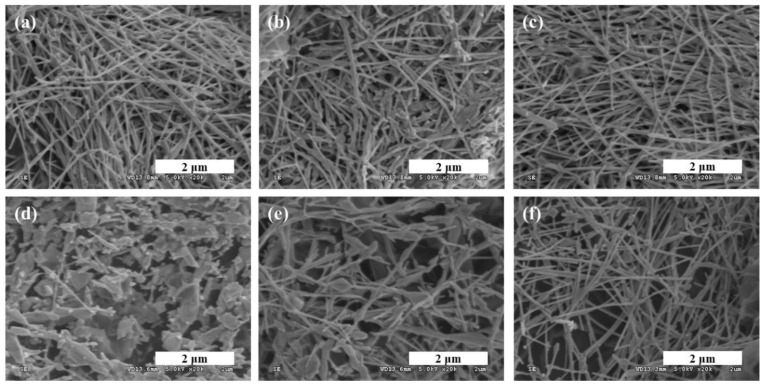
Surface scanning electron microscopy (SEM) images of the electrodes before (top) and after (bottom) a driving test of IEAPs based on (**a**,**d**) GO–Ag NWs, (**b**,**e**) P/GO–Ag NWs, and (**c**,**f**) TP/GO–Ag NWs.

**Table 1 polymers-11-00906-t001:** Sheet resistances before and after methanol treatment for various Triton X-100 weight ratios.

Samples	Sheet Resistance (mΩ/sq.)	
	Before Methanol Treatment	After Methanol Treatment
GO–Ag NWs	352.65	320.25
0.0 wt % (Pure PEDOT:PSS)	319.36	284.16
1.0 wt % Triton X-100	255.52	214.73
2.5 wt % Triton X-100	232.15	195.57
5.0 wt % Triton X-100	213.80	183.57
7.5 wt % Triton X-100	200.08	161.00
10.0 wt % Triton X-100	201.00	174.00

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
