# Peer review of "Enhanced Stability and Driving Performance of GO–Ag-NW-based Ionic Electroactive Polymer Actuators with Triton X-100-PEDOT:PSS Nanofibrils"

_polymers, 2019, doi:10.3390/polym11050906_

Round 1

Reviewer 1 Report

A Manuscript entitled "Enhanced stability and driving performance of GO-Ag NWs based ionic electro-active polymer actuators with Triton X-100-PEDOT:PSS nanofibrils" is about interesting topic of preparation and characterization of new ionic EAP type artificial muscle.

During the reading of Manuscript arose the following suggestions, comments and questions:

1) English language and style needs to be improved.

2) The manuscript contains a number of errors that may result from incorrectly converting a text file to a PDF, such as:

a) line 36 in manuscript "... high energy density1 ..."

b) line 81 in manuscript "... Hummer's method.21 ..." 

c) line 91 in manuscript "... papers.19 ..."

2) The materials sub-section (line 82) states: "... 20 wt% Nafion resin ...". What is this? Is it actually Nafion solution?

3) The materials sub-section (line 82) states: "... were purchased from Dupont™. ..." It's probably impossible to buy anything from a DuPont TradeMark, but it's definitely possible to buy products from DuPont company.

4) All abbreviations must be written out for the first time, even in case of this abbreviation (line 87): "... DI water ... "

5) List of used aparatus is missing. Please also provide the information about manufacturer and model.

6) I did not find the dimensions of the studied actuator in the manuscript. Figure 5 is almost useless without this information.

7) The introductory section states that: "Ionic electro-active polymers (IEAPs) ... have ... a ‘sandwich’ structure of electrode/ionic polymer/electrode." This statement is not always true.

There are a large number of publications in which the polymer used to make the porous  membrane for actuator is not an ionic polymer, such as:

a) Fukushima T, Asaka K, Kosaka A and Aida T 2005 Angew. Chem. Int. Edn 44 2410-3

b) Mukai K, Asaka K, Kiyohara K, Sugino T, Takeuchi I, Fukushima T and Aida T 2008 Electrochim. Acta 53 5555-62

8) Why do you think you can wash out the polystyrene sulphonate? As far as I know, PEDOT is a polycationic polymer in an oxidized state, and its charge is compensated by polystyrene sulfonate, which is a polyanionic polymer. Thus, Triton X-100, a non-ionic surfactant, should not allow to wash out  polystyrene sulfonate with methanol. Have you somehow confirmed the polystyrene sulphonate wash out? Please comment.

Author Response

Dear Editor and referees 

  Thank you for your reviews and suggestions. We revised our manuscript according to reviewers’ comments and suggestions. Then, the manuscript was sent to a professional company for proof reading to check the English grammar and descriptions.

Reviewer 2 Report

The paper by Park et al. describes the investigation of stability and driving performance of modified GO-AgNWs. This topic is very interesting and most of the science appears well done but before the paper can be published in Polymers some modifications should be done:

1.       Some brackets at references in the text are missing (see e.g. lines 36, 81, 91)

2.   Figure 1b shows the optical transmittance of glass slides coated by Triton/PEDOT:PSS mixtures. It is known that the absorbance/transmittance of the layer is dependent on the thickness of the layer. In my opinion the comparison of the transmittances of different layers would make sense only when there is a comparable thickness of layers. Profilometry measurements confirming the thickness of the films should be provided and the value of the average thicknesses should be added.

Author Response

(The authors gave the same response as above.)

Round 2

Reviewer 1 Report

The quality of the manuscript has improved.

I recommend to accept the manuscript in its current form.

Reviewer 2 Report

All the questions were addressed. Therefore, the present version of the manuscript can be accepted by the journal.